# Exploring the Impact of the *NULL* Class on In-the-Wild Human Activity Recognition

**DOI:** 10.3390/s24123898

**Published:** 2024-06-16

**Authors:** Josh Cherian, Samantha Ray, Paul Taele, Jung In Koh, Tracy Hammond

**Affiliations:** Department of Computer Science & Engineering, Texas A&M University, College Station, TX 77843, USA; sjr45@tamu.edu (S.R.); ptaele@tamu.edu (P.T.); jungin@tamu.edu (J.I.K.)

**Keywords:** human activity recognition, activities of daily living, class imbalance, preprocessing, postprocessing, in-the-wild, smartwatch

## Abstract

Monitoring activities of daily living (ADLs) plays an important role in measuring and responding to a person’s ability to manage their basic physical needs. Effective recognition systems for monitoring ADLs must successfully recognize naturalistic activities that also realistically occur at infrequent intervals. However, existing systems primarily focus on either recognizing more separable, controlled activity types or are trained on balanced datasets where activities occur more frequently. In our work, we investigate the challenges associated with applying machine learning to an imbalanced dataset collected from a fully *in-the-wild* environment. This analysis shows that the combination of preprocessing techniques to increase recall and postprocessing techniques to increase precision can result in more desirable models for tasks such as ADL monitoring. In a user-independent evaluation using *in-the-wild* data, these techniques resulted in a model that achieved an event-based F1-score of over 0.9 for brushing teeth, combing hair, walking, and washing hands. This work tackles fundamental challenges in machine learning that will need to be addressed in order for these systems to be deployed and reliably work in the real world.

## 1. Introduction

Human activity recognition has been a significant focus of research and development in wearable and ubiquitous computing for the last several decades due to its ability to provide a real-time understanding of human behavior and its potential to inform intelligent and personalized user-centered technologies. This recognition has been powered by a combination of sensors, located either on the body or in the environment, and machine learning techniques that have become increasingly adept at distinguishing among a variety of human behaviors and activities. Researchers have applied human activity recognition techniques to a diverse range of applications, including healthcare and well-being [1,2,3,4], weightlifting and sports [5,6,7,8], sign language translation [9], and car manufacturing and safety [10,11]. Within the area of healthcare and well-being, researchers have devoted particular attention to the recognition of activities of daily living (ADLs), as ADL performance is a key indicator of day-to-day health and wellness [12,13,14]. Over the years, researchers have developed pipelines capable of recognizing a handful of ADLs [15,16,17,18,19,20,21] or specific ADLs of interest, such as washing hands [22,23,24,25,26], taking medication [2,27], brushing teeth [28], and eating and drinking [29,30,31,32,33,34]. Such systems could be beneficial in practice to a number of populations: parents could guarantee that their children are learning and maintaining good health habits, caregivers could ensure that older adults are safely and successfully taking care of themselves, and the average person could track how well they are maintaining their day-to-day health. Furthermore, they would provide flexibility and facilitate greater autonomy than the existing paradigm for monitoring day-to-day health.

However, for these systems to be widely adopted by these populations, they need to be accurate, reliable, and robust in real-world settings. This requires these systems to be built and evaluated on data captured during the real-world performance of ADLs, as the performance of ADLs in controlled and semi-naturalistic settings often differs from the performance of ADLs in real-world settings [15,35,36]. Additionally, such systems must be able to differentiate between the performance of ADLs of interest and the performance of every other activity an individual performs. Often, ADLs of interest are a small minority of the activities performed on a given day. As a result, reliable human activity recognition in the wild requires overcoming a significant data imbalance issue. As examples, ADLs such as washing hands, taking medication, and brushing teeth, which are indicators of day-to-day health and hygiene, only occur at most a few times per day, with each instance only lasting seconds to a few minutes. Even within the broader field of machine learning, the issue of class imbalance remains a challenging open problem [37].

Although recent studies in the field of human activity recognition have acknowledged the challenge of in-the-wild recognition, the practicality of using many of the models proposed by human activity recognition studies in deployed systems remains limited by sub-optimal performance, the nature of the datasets that these models are trained on, or both [21,38,39,40]. The limitation of the nature of the datasets is perhaps the most endemic to the field, as the majority of studies rely on publicly available datasets that have a limited number of labeled activities and are comprised of data collected in controlled or semi-naturalistic environments. Thus, these models remain untested on data produced under real-world conditions (i.e., on *in-the-wild* data), which consist of more activities and feature a significantly higher level of class imbalance. The work by Vaizman et al. [41] highlights the importance of ensuring that models perform well *in real-world settings*, emphasizing that deployed applications need to work in a variety of contexts and when behaviors are performed irregularly. The more recent work by Bhattacharya et al. [21], seeking to evaluate their pipeline in real-world settings, evaluated an activity recognition pipeline on *in-the-wild* data. However, their work did not consider the large *NULL* class, i.e., all activities that are not of interest, which predominantly comprises *in-the-wild* data. The authors acknowledge this and note that the problem of *in-the-wild* ADL recognition requires more attention, as there is still significant room for performance improvement.

To address these limitations, we investigated the design of a human activity recognition system that has been trained on *in-the-wild* data in order to detect a set of ADLs. Specifically, we consider standard methods for handling imbalanced data, introduce a postprocessing technique to improve prediction precision, and assess the performance of classical feature-based models and deep learning models within these data-processing pipelines. These results establish a baseline for user-independent, *in-the-wild* activity recognition for a set of common ADLs. The main contributions of our work are as follows:**A fully** ***in-the-wild***** dataset.** First, we present an annotated *in-the-wild* dataset, which consists of accelerometer and gyroscope data from off-the-shelf smartwatches worn on both wrists. This dataset consists of 106.74 h of data from nine participants behaving naturally and following their personal daily routines in their homes and workplaces.**An evaluation of existing techniques to handle imbalances in human activity recognition data.** Second, we investigate techniques to improve classification performance on activity recognition systems trained on *in-the-wild* data. These techniques include common methods for dealing with imbalanced classes (e.g., undersampling and oversampling), as well as model training strategies such as cost-sensitive learning. Our experiments show that, in the case of *in-the-wild* data, these techniques improve the recall of the model at the cost of its precision. As a result, we find that these techniques in isolation are not enough to address the challenges associated with *in-the-wild* recognition.**A novel postprocessing technique.** Third, we propose a context-based prediction correction method to improve prediction stream stability. We evaluate the performance of this algorithm with five different weighting functions using the best-performing models from the previous experiments with and without preprocessing. Our model achieved an event-based F1-score of over 0.9 for the activities of brushing teeth, combing hair, walking, and washing hands in a user-independent evaluation using both preprocessing and postprocessing techniques.

## 2. Related Work

Given the impact that data imbalance and class overlap have on classification performance, both of these issues have received attention from researchers over the years. In this section, we provide an overview of some of the more common approaches that researchers have taken to handle these challenges and discuss publicly available human activity recognition datasets.

### 2.1. Human Activity Recognition

Researchers have been exploring ways of recognizing human activities for several decades, with early work exploring where sensors might be placed and what these form factors might look like [36,42,43]. As smartphones and wearable sensing devices such as fitness trackers and smartwatches have become increasingly ubiquitous and machine learning techniques have become increasingly advanced, researchers have shown that reliably recognizing many everyday human activities is technically achievable in broadly appealing form factors. As the field has been researched for so long, there are a number of papers that have surveyed the breadth of work that human activity recognition researchers have accomplished, covering the sensors used, common techniques and algorithms, applications, public datasets, and key challenges and future directions [38,44,45,46]. We refer interested readers to these works for a more in-depth understanding of the field.

### 2.2. General Techniques for Dealing with Data Imbalances

Techniques designed to handle data imbalances in machine learning applications are applied as a preprocessing step (data level), as part of the training of the model (algorithm level), or both (hybrid) [47,48,49]. Data-level techniques are methods designed to change the class distribution of an imbalanced dataset, allowing an algorithm to pay more attention to the minority class (or classes). As these techniques are applied prior to the application of the algorithm, they are commonly referred to as preprocessing techniques and generally utilize either undersampling or oversampling approaches. Undersampling approaches reduce the size of the majority class. Common undersampling techniques include Random Undersampling (RUS) [50], Condensed Nearest Neighbor [51], and Tomek Links (TL) [52]. Newer approaches have proposed various techniques for targeting specific data points within the majority class for removal [53,54]. Oversampling techniques increase the size of the minority class. The most popular oversampling approaches are Random Oversampling (ROS) and SMOTE [55], with many of the more recent oversampling techniques being modifications to the latter [56,57]. There have also been several works that both undersample the majority class and oversample the minority class(es) [58,59]. These data-level techniques are inherently flexible, as they can simply be inserted into the pipeline without requiring any significant modifications downstream. However, these techniques can over- or underemphasize patterns inherent to the data, leading to models that are less applicable to real-world scenarios.

Algorithm-level techniques refer to modified versions of existing algorithms that are designed to handle class imbalances. These techniques are generally designed for specific tasks and datasets, sacrificing flexibility for better performance on a narrower problem. By far the most common algorithm-level methods are cost-sensitive techniques in which the algorithm imposes a higher cost on misclassifying the minority class [60,61]. Other algorithm-level approaches include single-class learning, in which a classifier is trained on just the minority class, and boundary modification methods, which shift the decision boundary artificially to account for class imbalances [62,63]. Hybrid approaches seek to combine data-level and algorithm-level approaches in ways that maximize their benefits while mitigating their drawbacks. These approaches largely operate by utilizing undersampling or oversampling approaches to balance the data seen by weak learners during the training of the algorithm. Liu et al. [64] proposed EasyEnsemble and BalanceCascade, approaches that consist of an ensemble of classifiers trained on undersampled subsets of the data. Several studies have incorporated undersampling and oversampling into boosting, using RUS [65] and SMOTE [66] to resample the data prior to training the weak learner during each round of boosting, while the work by Wang and Yao [67] similarly evaluated the efficacy of RUS, SMOTE, and ROS to balance the data seen by the baseline estimators of a Bagging classifier.

### 2.3. Handling Data Imbalances in Human Activity Recognition

There are a number of works within the field of human activity recognition that have developed or utilized techniques specifically designed to mitigate the effects of data imbalances. At the data level, several works have employed existing undersampling and oversampling techniques [68,69]. Alharbi et al. [40] and Nguyen et al. [70] introduced novel oversampling methods, which they evaluated on several publicly available datasets. At the algorithm level, Guo et al. [39] developed a dual-ensemble framework, an algorithm-level technique that utilizes a nested ensemble structure comprised of an internal ensemble model that selects a base classifier from three classifiers based on classification performance on a balanced subset of the data and an external ensemble model that uses an evolutionary algorithm to find the optimal combination of base classifiers. Several works have utilized cost-sensitive learning, evaluating the efficacy of weighted extreme learning machines (ELMs) to handle data imbalances in publicly available datasets [71,72]. While these works have directly looked at imbalances, they all target the issue of data imbalances between classes of interest (e.g., in the Opportunity dataset, drinking from a cup occurs 22% of the time, while the next most frequent activity, cleaning a table, occurs 7% of the time). Conversely, our work focuses on the imbalance between classes of interest and the *NULL* class, which comprises all of the activities that a person performs that are not of interest. While both forms of imbalance are prevalent in real-world situations, addressing the latter form means developing techniques that are robust to significantly higher forms of imbalance. Even within the broader field of machine learning, most works have focused on evaluating techniques to handle imbalances in datasets with imbalance ratios ranging from 1:4 to 1:100 [73], while our work utilizes a dataset with a much higher level of imbalance.

## 3. Materials and Methods

### 3.1. Data Collection

We collected 3D accelerometer and gyroscope data from two Polar M600 (Polar Electro, Kempele, Finland) smartwatches worn by participants on both of their wrists [74]. The watches transmitted raw sensor data to Android smartphones in real time using a custom-built data collection application. Participants used this smartphone application to start and stop data collection and label the data being collected. The application had buttons for the following labels: “Washing Hands”, “Brushing Teeth”, “Eating”, “Drinking”, “Taking Meds”, “Combing Hair”, “Clapping”, “Walking”, “Nothing”, and “Other”. By default, the data were labeled as “Nothing”. When participants performed an activity, they selected the label for the activity and deselected the label (or selected “Nothing”) when they were done performing the activity. If the participant performed an activity that did not have a corresponding button and wanted to label that activity, pressing the Other button allowed them to type in another label. A diagram of the experimental setup is shown in Figure 1.

We recruited nine participants (aged 18–30 years; 2 females) to wear the sensors during their daily lives. Participants wore the devices as long as they wanted; the duration varied from 3 h to 34 h over the course of two days with the sensors off during the night. Four of the nine participants participated in a second data collection session over a year after the first data collection session. The data collection protocols were reviewed and approved by our organization’s Institutional Review Board (IRB). Cherian et al. [2] used these data in their work to recognize the activity of taking medication.

### 3.2. Activities

In this work, we sought to recognize the following seven activities: brushing teeth, combing hair, drinking, eating, taking medication, walking, and washing hands. Thus, data labeled as “Clapping” were not used in this study. Additionally, any activities labeled by participants using the “Other” button were relabeled as “nothing” for our analysis, with the exception of ascending and descending stairs, which were relabeled as walking. Additionally, one participant used an electric toothbrush when brushing their teeth; these data were relabeled as “nothing”, as using an electric toothbrush does not require the use of the same physical movements as using a manual toothbrush.

Within the field of human activity recognition, the *NULL* class is commonly used to collectively describe unknown or out-of-scope classes [21]. Thus, throughout the remainder of this work, we refer to all data within our dataset that were not labeled as one of the seven activities (i.e., labeled as “nothing”) as the *NULL* class.

### 3.3. Dataset Metrics

As expected in a real-world study, the majority of the data we collected in our studies were not samples of the ADLs of interest, reflecting the reality that these are relatively short activities that are only performed a few times over the course of a day. In total, we collected 106.68 h of sensor data, and only 5.84 h or 5.47% of the data were samples of ADLs. Table 1 gives the breakdown of data by class in terms of hours and imbalance.

For imbalance, we calculated the imbalance ratio (*IR*), which is a simple ratio between the number of data points comprising the majority class and the number of data points comprising the minority class. This definition is logical for binary classification tasks, but it loses detail for multiclass classification where minority classes are not the same size. In this work, we report the average *IR* across activity classes using Equation (Equation 1). Here, $\left|C_{a}\right|$ is the number of instances belonging to class *a*, *n* is the total number of instances including the *NULL* class, and *k* is the number of activity classes.
(1)$$IR_{avg}=\frac{\left(\sum_{a=1}^{k}\frac{n-|C_{a}|}{|C_{a}|}\right)}{k}$$

### 3.4. Activity Recognition Performance Metrics

Human activity recognition studies commonly report performance on the classification of short windows of sensor data using one or more of the following metrics: accuracy, sensitivity, specificity, precision, recall, and F1-score [46]. Of these metrics, the precision, recall, and F1-score are especially popular among studies that evaluate their systems on datasets with some level of imbalance. However, these metrics generally quantify a system’s ability to recognize short characteristic segments or windows of sensor data, whereas the success of a real-world implementation of such a system depends on its ability to recognize larger activity events. That is to say, the event-level performance, rather than the window-level performance, is a more relevant indicator of a system’s practical viability.

Thus, in this work, we report the performance at both the *window level* and the *event level*. For the latter, we utilize the performance metrics put forth by Ward et al. [75], who define an event as a “variable duration sequence of positive frames within a continuous time-series” that “has a start time and a stop time”. At the event level, Ward et al. defined several event categories, which are defined in Figure 2. Based on these event categories, we define the event-level precision (Equation (Equation 2)), event-level recall (Equation (Equation 3)), and event-level F1-score (Equation (Equation 4)).
(2)$$P_{event}=\frac{C+M^{\prime }+FM^{\prime }+F^{\prime }}{C+M^{\prime }+FM^{\prime }+F^{\prime }+I^{\prime }}$$
(3)$$R_{event}=\frac{C+F+FM+M}{C+F+FM+M+D}$$
(4)$$F1_{event}=2\frac{P_{event}\cdot R_{event}}{P_{event}+R_{event}}$$

### 3.5. Algorithms

We evaluated the efficacy of both classical machine learning algorithms and deep learning models in recognizing the performance of ADLs in real-world scenarios.

#### 3.5.1. Classical Machine Learning Models

Table 2 gives an overview of the features used in the classical machine learning algorithms. These features were extracted using a one-second sliding window with 75% overlap. Features (A) through (Q) were calculated for each axis of the accelerometer for each hand, and features (R) and (S) were calculated for each hand, bringing the total number to 106 ((17×3axes×2hands)+(2×2hands)).

Most of the included features are statistical or physical characteristics of a signal and are commonly used in activity recognition [2,15,21,28,76,77]. Equations (Equation 5)–(Equation 7) are used to calculate (O)–(Q) [78], where $X_{i}\left(k\right)$ represents the amplitude of the *k*th bin of the DFT performed on the sensor signal. Equation (Equation 8) is used to calculate (S) [79], where N(ϵ) is the number of boxes of length ϵ required to cover the sensor signal *A*.
(5)$$C=\frac{\sum_{k=1}^{N}kX\left(k\right)}{\sum_{k=1}^{N}X\left(k\right)}$$
(6)$$S=\sqrt{\frac{\sum_{k=1}^{N}{\left(k-C\right)}^{2}X\left(k\right)}{\sum_{k=1}^{N}X\left(k\right)}}$$
(7)$$\underset{SR}{\arg\;\min}\;\sum_{k=1}^{SR}X\left(k\right)\geq 0.85\sum_{k}X\left(k\right)$$
(8)$$D\left(A\right):=\lim_{\epsilon \to 0}\frac{\log N(\epsilon )}{\log(1/\epsilon )}$$

In this work, we tested Naive Bayes, Random Forest (100 trees, max_depth = 30), SVM (RBF kernel), and XGBoost (version 1.7.1) (100 trees, max_depth = 30). These models were implemented with sklearn (version 1.1.1) [80], with the exception of XGBoost, which was implemented using the xgboost python package developed based on work by Chen and Guestrin [81].

#### 3.5.2. Deep Learning Models

In this work, we evaluated two deep learning models: DeepConvLSTM [82] and Attend&Discriminate [83]. The former is a deep learning network comprised of convolutional and LSTM recurrent layers and has been used as a benchmark in several recent studies [21,84]. The latter uses a self-attention mechanism to learn the relations between multiple sensor channels and a temporal attention mechanism to learn the importance of individual time steps. Both of these models were implemented using PyTorch (version 1.12.1) [85] and use designs that align with prior work [21]. In this work, the DeepConvLSTM processes accelerometer and gyroscope features separately through its own convolutional layers and LSTMs before concatenating the resulting features into a final classifier. For both types of features, we use four convolutional layers with 64 filters and a linear kernel of size 5. Batch normalization occurs after each convolution, and max pooling occurs every two convolutions with a linear kernel of size 3. The Attend&Discriminate model uses two convolutional layers with 8 filters and a linear kernel of size 3. Self-attention is applied along each sensor channel before feeding into a GRU and then a final classifier.

### 3.6. Existing Methods for Addressing Imbalances

  As discussed in Section 2.3, researchers have suggested a number of techniques for handling imbalances. In this work, we evaluate several widely used methods at both the data level and algorithm level, as our aim is to provide insight into the ability of common techniques to mitigate the effects of the imbalances characteristic of real-world human activity recognition. Specifically, we look at random undersampling (RUS), random oversampling (ROS), and cost-sensitive learning (CSL). In RUS, the majority class is resampled without replacement such that it is the same size as the non-majority samples. In ROS, random samples from the minority class are duplicated to decrease the data skew. Generally, this practice entails oversampling each minority class such that it is equal in size to the majority class. Due to the extreme imbalance ratios for certain classes, however, balancing the classes would result in a poor model that has the a priori assumption that all classes occur with equal frequency. Thus, instead, we oversampled our minority classes such that the ratios between the minority classes remained the same and the total number of non-majority samples equaled the number of majority samples. Both undersampling and oversampling were implemented using the Imbalanced-learn python package [86]. For cost-sensitive learning, which occurs at the classifier level, we used balanced class weights, i.e., each class was weighted inversely proportionally to its size.

### 3.7. Classification Postprocessing

To improve the stability of the prediction stream, we take advantage of the temporal nature of the problem of activity recognition. Specifically, we leverage the basic truth that a data point is most likely to have the same class as the data points that immediately precede and succeed it. Past works in the field of activity recognition have taken advantage of this idea; for instance, the work by Cherian et al. [28] utilized a simple set of rules specific to the activity of brushing teeth to improve their algorithm’s ability to identify instances of that activity. Other works use simple rules such as taking the majority of a group of predictions after a delay [87,88,89,90]. In this work, we take a more general approach to avoid the necessity of having to develop subjective activity-specific rules.

Intuitively, our approach corrects predictions by considering preceding and succeeding predictions and giving higher weights to neighboring predictions that are closer to the prediction under consideration. Given an array of predicted labels $P=[P_{1},\cdots ,P_{n}]$, where *n* is the number of windows extracted from the raw data stream, a corrected array of predictions $P^{\prime }$ can be generated by analyzing the preceding and succeeding predictions of each prediction $P_{i}$. The subset of predictions centered on $P_{i}$, which we denote as *W*, is used to update $P_{i}$ to the most likely prediction given this surrounding context using a weighting function. Algorithm 1 illustrates our approach at the highest level, showing how the stream of predictions *P* is converted into a corrected stream of predictions $P^{\prime }$. Algorithm 2 demonstrates how the predictions surrounding $P_{i}$ are weighted and used to produce the corrected prediction. We investigated several different weighting functions; these are given in Table 3. Note that in deployment, this technique will introduce a latency to the classification determination that is proportional to the number of predictions following $P_{i}$ that are considered as part of *W*.
**Algorithm 1:** Classification Postprocessing Algorithm**Input**: *P*: prediction stream, $w_{s}$: context window size, weight: context window scoring function**Output**: $P^{\prime }$: corrected prediction stream for *P*1$P^{\prime }\leftarrow \left[\right]$2W←[]3**for** *i in (1 to $\left|P\right|$)***:**4 **if** *$\left|W\right|$ < $w_{s}$*5  Push P[i] onto $P^{\prime }$6 **else**7  Push postprocess (*W*, $w_{s}$, weight) onto $P^{\prime }$8  Pop front *W*9 Push P[i] onto *W*10**return** $P^{\prime }$
**Algorithm 2:** Context-basedPrediction Correction**Input**: *W*: context window, $w_{s}$: context window size, weight: context window scoring function**Output**: *p*: prediction1counts←Counter(W)2scores←map()3**for** *x in counts.keys***:**4  scores[x]←05$midpoint\leftarrow \lfloor \frac{\left|W\right|}{2}\rfloor +1$6**for** *i in (1 to $\left|W\right|$)***:**7  $scores\left[W\left[i\right]\right]\leftarrow scores\left[W\left[i\right]\right]+weight\left(w_{s},i-midpoint+1\right)$8p←counts.keys[0]9max←010**for** *x in counts.keys***:**11 **if** *scores[x] > max*12  max←scores[x]13  p←x14 **else if** scores[x]==max15  **if** counts[x]>counts[p]16   p←x17**return** *p*

## 4. Results

While most works in human activity recognition utilize some form of cross-validation, given our dataset’s size, we evaluated our pipeline’s efficacy using a training–validation–evaluation split on a 70%, 20%, and 10% basis. For these splits, we used a stratified approach on the participant level to ensure that ADL data were distributed as evenly as possible among datasets and that data streams were contiguous. This design also results in a user-independent evaluation.

### 4.1. Impact of Preprocessing Techniques

We present the results of three common preprocessing techniques using the definitions in Section 3.6 on the validation set to demonstrate the impact on model classification performance alongside the baseline performance of each model. We show the macro precision, macro recall, macro F1-score, event-level precision, event-level recall, and event-level F1-score in Table 4. We report the macro definitions instead of the weighted definitions, as data imbalances inflate weighted performance metrics. Naive Bayes is not included in the cost-sensitive learning trial, as this model does not have a concept of class weights.

Of the models and configurations tested, XGBoost with no preprocessing performed the best, achieving an event-level F1-score of 0.52. XGBoost with ROS achieved the second-highest performance, with an event-level F1-score of 0.50. Looking at the results overall, preprocessing techniques generally led to decreases in performance; although they largely improved the event-level recall, they also caused decreases in the event-level precision. It is worth noting that Naive Bayes consistently achieved near-perfect event-level recall; however, this was always offset by abysmal event-level precision.

### 4.2. Impact of Postprocessing

We evaluated our postprocessing technique on the highest-performing classifiers with and without preprocessing from the previous experiment: baseline XGBoost and XGBoost with ROS. We investigated all weighting functions given in Table 3 with window (*W*) sizes that ranged from 10 s to 240 s in increments of 10 s. Precision and recall tended to increase as the window size grew up to a plateau point; after that point, recall remained stable, while precision started to decrease, lowering the overall F1-score. Table 5 shows the highest-performing postprocessing algorithm for the baseline and preprocessing conditions for the validation set alongside the original performance metrics for direct reference. Overall, XGBoost with ROS and our postprocessing algorithm achieved the highest results with an event-level F1 score of 0.64, an improvement of 0.12 from the highest-performing model without any postprocessing.

In contrast to the effects of the preprocessing techniques, postprocessing increased the system’s precision at the cost of its recall. Furthermore, the effects of preprocessing and postprocessing did not cancel each other out, as they resulted in net gains for both precision and recall and produced the best overall pipeline for *in-the-wild* ADL recognition.

### 4.3. Evaluation

We present the results of the highest-performing models from the preprocessing and postprocessing experiments on the held-out evaluation set. Performance metrics are given in Table 6, and confusion matrices are given in Figure 3. While the gains were not as high as those on the validation set, the evaluation set saw improvements in the event-level F1-score performance metric. Additionally, the confusion matrices demonstrate an increase in model specificity with a decrease in false positives on the ADL classes. The effect of postprocessing on event-based performance metrics per class is given in Table 7. The model was not able to detect the classes of drinking or taking medication but was able to recognize the other classes to varying extents. Otherwise, the addition of postprocessing improved the precision without lowering the recall. Notably, of the classes that the model was able to recognize, the activities of brushing teeth, combing hair, and washing hands achieved an event-level F1-score of at least 0.97, and the activity of walking achieved an event-level F1-score of 0.91.

## 5. Discussion

### 5.1. Challenge of Non-Distinct Minority Classes

Recognizing activities performed in real-world scenarios is both the ultimate goal of human activity recognition and the most difficult version of this classification task. Our baseline results show that both traditional and deep learning models, even when combined with preprocessing steps, struggle to detect some of the classes. This outcome implies that these ADLs are not distinct from the *NULL* class. In general, imbalances hinder the recall of minority classes, which is problematic when the goal of the activity recognition model is to detect instances of infrequent behaviors such as ADLs. Techniques for handling imbalances, such as data preprocessing or cost-sensitive learning, can improve the recall for these classes of interest. However, the impact of these techniques creates unintended consequences for the performance of the model. By learning decision boundaries that favor the detection of these minority classes, models will incur false positives on nearby samples that belong to other classes. In this case, it is to be expected that gains in recall will be counterbalanced by equal if not worse losses of precision. We observe this phenomenon in the first experiment, where we investigate the impacts of preprocessing; the recall improves, but the overall performance in terms of the F1-score decreases. However, while preprocessing alone does not improve performance, it does synergize with our proposed postprocessing technique. We designed our postprocessing technique to correct the prediction stream based on the local context, allowing it to intuitively clean up false positives and false negatives. Because of the imbalances in the dataset and the nature of the data themselves, predicted events tend to be small in size. As such, postprocessing can convert true positives into false negatives when a true event is detected with low coverage. However, because preprocessing increases recall, these true events become less likely to be incorrectly adjusted, and incorrect detections will receive the same treatment as true detections without preprocessing. In short, preprocessing and postprocessing contribute to the overall performance of the model more than they hinder it, and their respective benefits cancel out the other’s costs.

### 5.2. Intraclass Variability in Certain ADLs

The nature of certain ADLs makes them difficult to detect in *in-the-wild* data, especially in user-independent evaluation. Exemplifying this concept, the final evaluation model was unable to detect the activities of drinking or taking medication even after using preprocessing. While it was detected, performance on the eating ADL was notably lower than that on the remaining ADLs. One attribute that these activities have in common is that they all involve consuming various substances. The more telling similarity for why their performance was low, however, is the number of ways these activities can be performed. These activities can involve one or two hands, which can be used sequentially or simultaneously to complete the activity, resulting in significant intraclass variability. The participants ate different foods with or without utensils, opened different types of packaging when taking medication, and used different styles of cups or mugs for drinking. Due to a lack of similar samples in the training set, these ADLs were commonly missed as part of the *NULL* class. Future work can focus on these activities to determine techniques for addressing this intraclass variability inherent to several ADLs.

### 5.3. Towards Real-World Human Activity Recognition

As with the real-world deployment of any machine learning system, the deployment of a human activity recognition system requires making a decision on which type of error is more unacceptable. In this work, we developed a postprocessing technique that was able to improve the overall performance of the system by dramatically increasing the precision at the cost of a decrease in the system’s recall. In other words, we decreased the number of false positives at the cost of an increase in the number of false negatives. For an application of human activity recognition such as elderly care (i.e., where a caregiver could remotely monitor the ADL performance of an older adult), this is likely the preferred choice, as this system would be highly certain when an older adult is performing their ADLs at the cost of caregivers sometimes still having to manually check in. Naturally, if the recall of such a system was too low, caregivers would likely feel as though such a system was not really saving them too much work, as they would still have to manually check most of the time. In contrast, for an application of human activity recognition such as surveillance, this paradigm would likely not be preferred, as human monitors would rather have to manually check a false alarm than miss suspicious behavior indicative of malicious intent. In this use case, a system whose precision was too low would create an onerous amount of additional work.

### 5.4. Hardware Considerations

Regardless of the classification performance of a human activity recognition system in real-world situations, such a system is not likely to fail without equal attention being placed on the specific consumer hardware on which these algorithms will run. In this work, we utilized Polar M600 watches [74], Android smartwatches available in the United States that run Google’s Wear OS, an operating system that is used by a number of different popular (and more modern) smartwatches currently on the market. A few users in our study did find this particular watch cumbersome; thus, developing algorithms for sensors and OSs that are widely available will be essential to ensure that users can choose the watch that fits their own personal preferences. The other major hardware consideration is power consumption, as these algorithms will likely be running in the background at all times to detect activities as they occur. In our study, users reported that the watch lasted a full day (that is, from when they put it on in the morning to when they took it off at night). Users who wore the watch a second day did have to charge the watches overnight, as the batteries were low by the end of the first day.

### 5.5. Future Work

There are a number of areas in which future work is necessary for real-world human activity recognition systems to become accurate and ubiquitous. In this work, our models were designed to differentiate among seven activities of daily living and a *NULL* class based on accelerometer and gyroscope data from a Polar M600. Future work should look at a broader range of activities, as well as other sensors and sensor locations, specifically in real-world settings. Additionally, given the potential of this field to aid different populations with specific use cases (e.g., children, older adults), future work should collect data from these populations and explore how solutions for handling the *NULL* class need to be adapted for these populations. Finally, in this work, we conducted a day-long study; future work can utilize data from longitudinal studies, exploring how to build models on weeks and months of data and how those models might be deployed and maintained effectively in real-world settings.

## 6. Conclusions

Recognition of human activities in a real-world environment is a difficult pattern recognition task. Over the course of a day, people perform an abundance of activities and motions, most of which will not be of importance to a human activity recognition system. As a natural consequence, these systems have trouble distinguishing uncommon activities from the swaths of data collected. In an effort to address this problem, we looked specifically at the recognition of human activities in real-world environments.

Our first contribution was collecting a fully *in-the-wild* dataset, in which the real-world performance of activities of daily living was annotated. The data were collected on commodity smartwatches on a diverse and representative set of activities that researchers can utilize to build richer human activity recognition systems. Our second contribution was investigating and discovering a novel postprocessing technique that improves classification performance for human activity recognition systems trained on *in-the-wild* data. The technique addresses the challenge of overlapping classes that researchers can utilize to build more robust human recognition systems. Our third contribution was our novel investigation directly into the class imbalance and class overlap problems when applying standard algorithms and data preprocessing techniques. This investigation can encourage other researchers in the domain of human activity recognition to expand our investigation into this open problem. These contributions collectively represent a significant step towards practical real-world recognition of human activities such as ADLs.

## Figures and Tables

**Figure 1 sensors-24-03898-f001:**
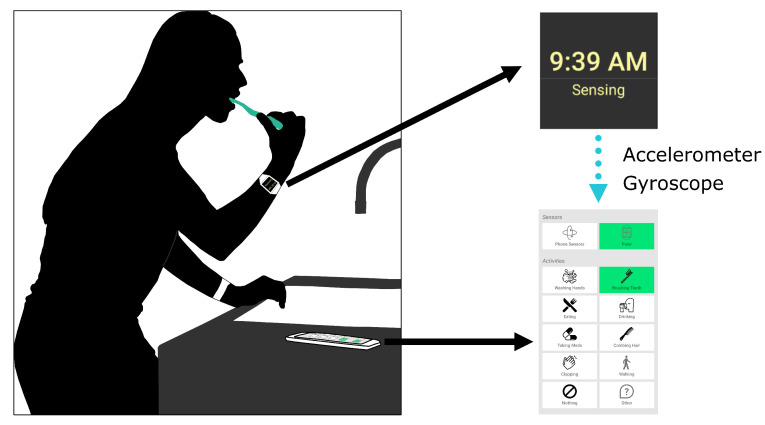
Data collection setup. Individuals were asked to wear watches on both wrists during data collection and use a custom-built data collection app on a provided smartphone to label activities of interest.

**Figure 2 sensors-24-03898-f002:**
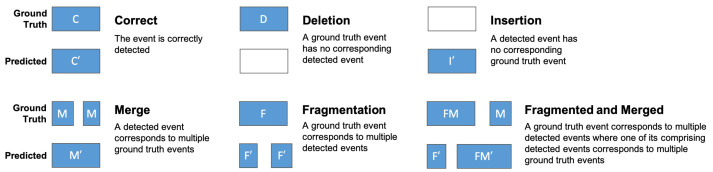
Visual definitions of the event categories that are used to calculate the event-level performance metrics. These were defined by Ward et al. [75].

**Figure 3 sensors-24-03898-f003:**
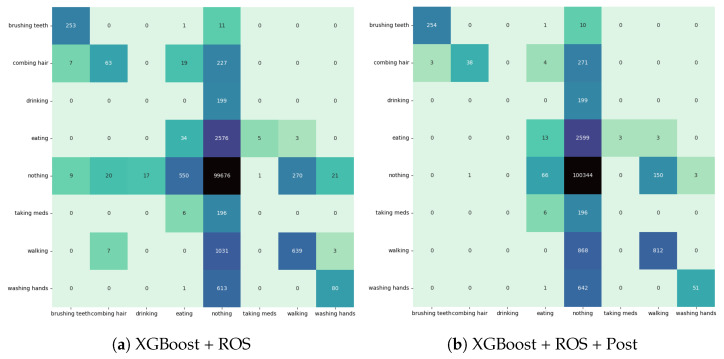
Confusion matrices of XGBoost’s performance on the evaluation set. The colors of the cells correspond to the frequency of the prediction outcome.

**Table 1 sensors-24-03898-t001:** Data size and imbalance ratios by class within the complete dataset. The overall IR and $R_{aug}$ are the average values across classes.

Activity Label		Total Data Size (H)	IR
BT	Brushing Teeth	0.24	500.72
CH	Combing Hair	0.23	457.15
DR	Drinking	0.21	509.85
EA	Eating	2.13	48.83
NU	NULL	100.90	-
TM	Taking Medication	0.29	364.25
WA	Walking	2.32	44.78
WH	Washing Hands	0.44	239.08
-	Overall	106.74	309.23

**Table 2 sensors-24-03898-t002:** Classical Machine Learning Features.

#	Feature Name		
(A)	Average Jerk	(K)	Standard Deviation of the Number of Peaks
(B)	Average Height	(L)	Number of Valleys
(C)	Standard Deviation Height	(M)	Average Number of Valleys
(D)	Energy	(N)	Standard Deviation of the Number of Valleys
(E)	Entropy	(O)	Spectral Centroid
(F)	Average	(P)	Spectral Spread
(G)	Standard Deviation	(Q)	Spectral Rolloff (<85%)
(H)	Root-Mean-Square	(R)	Axis Overlap
(I)	Number of Peaks	(S)	Fractal Dimension
(J)	Average Number of Peaks		

**Table 3 sensors-24-03898-t003:** Examples of weighting functions, where *d* is the distance from the window center.

#	Type	Formula
W1	Normal	$\lceil \frac{w_{s}}{2}\rceil -d$
W2	Normal Inverted	$\frac{1}{d+2}$
W3	Squared	${\left(\left\lceil \frac{w_{s}}{2}\right\rceil -d\right)}^{2}$
W4	Log	$\log(\left\lceil \frac{w_{s}}{2}\right\rceil -d+1)$
W5	Log Inverted	$\frac{1}{\log(d+2)}$

**Table 4 sensors-24-03898-t004:** Performance on the validation set. The model with the highest event-level F1-score is highlighted in bold.

Trial	Algorithm	P_*Macro*_	R_*Macro*_	F1_*Macro*_	P_*event*_	R_*event*_	F1_*event*_
Baseline	Naive Bayes	0.22	0.54	0.15	0.11	0.99	0.16
	SVM	0.23	0.36	0.23	0.13	0.51	0.17
	Random Forest	0.61	0.25	0.30	0.56	0.57	0.51
	**XGBoost**	**0.52**	**0.29**	**0.34**	**0.47**	**0.69**	**0.52**
	DeepConvLSTM	0.36	0.43	0.36	0.28	0.89	0.36
	Attend&Discriminate	0.37	0.38	0.36	0.29	0.84	0.39
RUS	Naive Bayes	0.22	0.56	0.18	0.11	1.00	0.17
	SVM	0.25	0.47	0.29	0.15	0.65	0.22
	Random Forest	0.31	0.51	0.34	0.22	0.82	0.30
	XGBoost	0.30	0.52	0.34	0.20	0.90	0.29
	DeepConvLSTM	0.25	0.53	0.28	0.14	0.89	0.21
	Attend&Discriminate	0.25	0.48	0.29	0.15	0.86	0.23
ROS	Naive Bayes	0.22	0.53	0.14	0.11	0.99	0.16
	SVM	0.19	0.46	0.19	0.07	0.90	0.12
	Random Forest	0.51	0.30	0.35	0.45	0.68	0.49
	XGBoost	0.46	0.33	0.36	0.40	0.77	0.50
	DeepConvLSTM	0.31	0.42	0.33	0.22	0.93	0.31
	Attend&Discriminate	0.24	0.55	0.28	0.13	0.97	0.21
CSL	Naive Bayes	-	-	-	-	-	-
	SVM	0.06	0.60	0.10	0.07	0.99	0.13
	Random Forest	0.41	0.35	0.36	0.34	0.76	0.43
	XGBoost	0.38	0.37	0.36	0.30	0.82	0.40
	DeepConvLSTM	0.27	0.53	0.31	0.17	0.95	0.24
	Attend&Discriminate	0.24	0.63	0.26	0.13	0.98	0.20

**Table 5 sensors-24-03898-t005:** Performance of the XGBoost classifier on the validation set with postprocessing.

Trial	Postprocessing	P_*Macro*_	R_*Macro*_	F1_*Macro*_	P_*event*_	R_*event*_	F1_*event*_
Baseline	-	0.52	0.29	0.34	0.47	0.69	0.52
Baseline	W4, 160 s	0.72	0.29	0.35	0.69	0.65	0.60
ROS	-	0.46	0.33	0.36	0.40	0.77	0.50
ROS	W4, 170 s	0.71	0.33	0.39	0.68	0.74	0.64

**Table 6 sensors-24-03898-t006:** Performance on the evaluation set.

Trial	Postprocessing	P_*Macro*_	R_*Macro*_	F1_*Macro*_	P_*event*_	R_*event*_	F1_*event*_
Baseline	-	0.55	0.31	0.35	0.49	0.63	0.53
Baseline	W4, 160 s	0.60	0.30	0.33	0.55	0.61	0.56
ROS	-	0.51	0.33	0.37	0.45	0.63	0.51
ROS	W4, 170 s	0.61	0.33	0.37	0.56	0.63	0.58

**Table 7 sensors-24-03898-t007:** Effect of postprocessing on the evaluation set by class for XGBoost with ROS. The performance of drinking and taking medication is not shown here, as the event-level precision, event-level recall, and event-level F1-score were all 0. F1-scores above 0.9 are highlighted in bold.

Activity	P_*event*_		R_*event*_		F1_*event*_	
	−	**+Post**	−	**+Post**	−	**+Post**
Brushing Teeth	0.94	0.98	1.00	1.00	**0.97**	**0.99**
Combing Hair	0.70	0.97	1.00	1.00	0.82	**0.99**
Drinking	0.00	0.00	0.00	0.00	0.00	0.00
Eating	0.06	0.14	0.43	0.43	0.10	0.21
Taking Medication	0.00	0.00	0.00	0.00	0.00	0.00
Walking	0.70	0.84	1.00	1.00	0.82	**0.91**
Washing Hands	0.77	0.94	1.00	1.00	0.87	**0.97**

## Data Availability

The raw data supporting the conclusions of this article will be made available by the authors upon request.

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
