# Peer review of "Exploring the Impact of the NULL Class on In-the-Wild Human Activity Recognition"

_sensors, 2024, doi:10.3390/s24123898_

Round 1
Reviewer 1 Report
Comments and Suggestions for Authors
I was interested in the paper's discussion of daily monitoring using smartwatches and the application of machine learning using the collected results. Unfortunately, I am curious about what data was collected through the smartwatch. Also, too much of it is not visualized, but expressed in text and tables. Since what is shown in the picture is almost a table, I would like to make the following suggestions.
1. Please draw a picture of what data you collected, for example, what numbers you recorded for the specific activities shown in Table 1 of the human activity recordings throughout the paper.
2. Explain specifically how the NULL state is defined.
3. Please also explain in detail how you distinguish between the different activities in Table 1.
I would like to come back to the paper after these things.
Author Response
Thank you for taking the time to review our manuscript and provide us with feedback. We have taken your suggestions into account; below we go into the changes we made to the manuscript and where they can be found.
1. "Please draw a picture of what data you collected, for example, what numbers you recorded for the specific activities shown in Table 1 of the human activity recordings throughout the paper."
We collected accelerometer and gyroscope data from a Polar M600 smartwatch. We have added Figure 1 on page 5 of the revised manuscript to better illustrate what data collection looks like.
2. "Explain specifically how the NULL state is defined."
The NULL class is a term used in human activity recognition to describe out-of-scope or unknown classes (as noted in reference 21). We have moved and expanded the definition of the NULL state; it can be found in Section 3.2 on lines 203-206 of the revised manuscript.
3. Please also explain in detail how you distinguish between the different activities in Table 1.
Activities were distinguished by user-generated labels. During data collection, participants were asked to label activities using the Android smartphone application. The expanded explanation of this can be found in Section 3.1 on lines 178-186 of the revised manuscript.
Reviewer 2 Report
Comments and Suggestions for Authors
The paper addresses one of the biggest challenges in Human Activity Recognition: dealing with imbalanced datasets. The combination of preprocessing and postprocessing techniques proposed to handle class imbalance and improve model performance is well-executed and shows high potential for practical applications. While the paper is commendable and makes a substantial contribution to the field, a minor revision is suggested to further strengthen the paper. Specifically, the paper would benefit from a more detailed description of the data collection process. Additionally, Is the data annotation completely reliant on user input?
Comments on the Quality of English LanguageThe writing is good and easy to understand.
Author Response
Thank you for taking the time to review our manuscript and provide us with feedback.
We have revised Sections 3.1 and 3.2 (pages 4-5) to include a more detailed description of the data collection process.
Yes, the data annotation is completely reliant on user input.
Reviewer 3 Report
Comments and Suggestions for Authors
This paper investigates the challenges associated with applying machine learning to an imbalanced dataset collected from a fully in-the-wild environment. To further improve the quality of the paper, the following suggestions are put forward:
1. This paper may consider comparing with existing literature on human activity recognition and the impact of the NULL class,so as to highlight the uniqueness and innovation of the paper work.
2. What type of 3D accelerometer and gyroscope are used in data collection in Section 3.1 of this paper, what is their measurement accuracy, and whether the data collected and transmitted through the application program is original data or pre-processed data. The presentation of this information may affect the credibility and repeatability of the paper.
3. The study only used a specific data set collected by the Polar M600 watch, which may limit the generality of the findings in other wearable devices or sensor configurations. If possible, it is recommended that data sets from multiple types of sensors or devices be used and analyzed, which will increase the reliability of the conclusions.
4. The lack of discussion of future work in the conclusion of the paper may affect readers' interest in the work.
Author Response
Thank you for taking the time to review our manuscript and provide us with feedback. Please find our responses below and the corresponding revisions in the updated manuscript.
1. "This paper may consider comparing with existing literature on human activity recognition and the impact of the NULL class, so as to highlight the uniqueness and innovation of the paper work."
This is a difficult comparison to make. As described in Section 2.3, there is some prior work in the field that has looked at data imbalance, but these have relied on publicly available datasets, where the imbalance is between the labeled classes and not between the classes and a large NULL class. Indeed there are no publicly available human activity recognition datasets with a realistically large NULL class, and so our work is a first step towards exploring this problem.
2. "What type of 3D accelerometer and gyroscope are used in data collection in Section 3.1 of this paper, what is their measurement accuracy, and whether the data collected and transmitted through the application program is original data or pre-processed data. The presentation of this information may affect the credibility and repeatability of the paper."
We used 3D accelerometer and gyroscope data collected from a Polar M600 smartwatch. We are not aware of what specific accelerometer and gyroscope is used within the watch or what their measurement accuracy is; to the best of our knowledge, this is not publicly available knowledge. The data collected and transmitted through the application program is original data (we have added the word “raw” on line 176 to make this explicit). We would like to note that data from the sensors within this watch has been used extensively in the field of human activity recognition (see references below); these papers do not indicate sensor type or accuracy either.
Akbari, Ali, et al. "Using intelligent personal annotations to improve human activity recognition for movements in natural environments." IEEE journal of biomedical and health informatics 24.9 (2020): 2639-2650.
Medina, Miguel Ángel López, et al. "Activity recognition for iot devices using fuzzy spatio-temporal features as environmental sensor fusion." Sensors (Basel, Switzerland) 19.16 (2019).
Akbari, Ali, Jonathan Martinez, and Roozbeh Jafari. "Facilitating human activity data annotation via context-aware change detection on smartwatches." ACM Transactions on Embedded Computing Systems (TECS) 20.2 (2021): 1-20.
Odhiambo, Chrisogonas O., et al. "Human activity recognition on time series accelerometer sensor data using LSTM recurrent neural networks." arXiv preprint arXiv:2206.07654 (2022).
3. "The study only used a specific data set collected by the Polar M600 watch, which may limit the generality of the findings in other wearable devices or sensor configurations. If possible, it is recommended that data sets from multiple types of sensors or devices be used and analyzed, which will increase the reliability of the conclusions."
Yes, this does limit the generalizability of our findings; however, using other sensors or devices is outside the scope of our current work. It’s worth noting here that the majority of human activity recognition papers that do not explicitly focus on topics such as sensor location or sensor fusion rely on data from a single device (especially in the case of studies that use a smartphone or smartwatch as a data source (see reference below)). Nevertheless, we have mentioned this in the Future Work section (Section 5.5).
Demrozi, Florenc, et al. "Human activity recognition using inertial, physiological and environmental sensors: A comprehensive survey." IEEE Access 8 (2020): 210816-210836.
4. "The lack of discussion of future work in the conclusion of the paper may affect readers' interest in the work."
We have added a Future Work section to our paper. This is now Section 5.5 and can be found on pgs 13-14.
Round 2
Reviewer 1 Report
Comments and Suggestions for Authors
I have verified that the authors have faithfully followed and corrected my comments. Therefore, I believe this draft is sufficient for publication in this paper.
Reviewer 3 Report
Comments and Suggestions for Authors
The authors have carefully modified the content of the paper as suggested previously. The research route of the paper is clearer and the research results are more credible.